# Traditional Ethnobotanical Knowledge of the Central Lika Region (Continental Croatia)—First Record of Edible Use of Fungus *Taphrina pruni*

**DOI:** 10.3390/plants11223133

**Published:** 2022-11-16

**Authors:** Ivana Vitasović-Kosić, Antonija Hodak, Łukasz Łuczaj, Mara Marić, Josip Juračak

**Affiliations:** 1University of Zagreb Faculty of Agriculture, 10000 Zagreb, Croatia; 2Institute of Biology and Biotechnology, University of Rzeszów, 36-100 Rzeszów, Poland; 3Department for Mediterranean Plants, University of Dubrovnik, 20000 Dubrovnik, Croatia

**Keywords:** ethnoveterinary tradition, rural area, wild edible plants, wild food plants, agrobiodiversity, ethnomycology, edible mushrooms

## Abstract

This study analyzed the use of plants and fungi, some wild and some cultivated, in three municipalities of Lika-Senj County (Perušić, Gospić and Lovinac). The range of the study area was about 60 km. Forty in-depth semi-structured interviews were performed. The use of 111 plant taxa from 50 plant families and five taxa of mushrooms and fungi belonging to five families was recorded (on average 27 taxa per interview). The results showed quite large differences between the three studied areas in terms of ethnobotanical and ecological knowledge. In the Perušić area, (101 taxa mentioned), some people still use wild plants on a daily basis for various purposes. The most commonly noted plants are *Prunus spinosa*, *Taraxacum* spp., *Rosa canina*, *Urtica dioica*, *Juglans regia* and *Fragaria vesca*. In the Lovinac region, people used fewer species of plants (76 species mentioned). The most common species used there are: *Rosa canina*, *Achillea millefolium*, *Cornus mas*, *Crataegus monogyna*, *Sambucus nigra* and *Prunus domestica*. In the town of Gospić, the collection and use of plants was not so widespread, with only 61 species mentioned, the most common being: *Achillea millefolium*, *Cornus mas*, *Sambucus nigra*, *Viola* sp., *Prunus domestica* and *Rosa canina*. The medicinal use of herbal tea *Rubus caesius* and *Cydonia oblonga* against diarrhea was well known in the study area and is used medicinally, mainly in the rural parts of the Gospić area. The consumption of the *Sorbus* species (*S. aria, S. domestica* and *S. torminalis*) is an interesting local tradition in Perušić and Lovinac. Species that are difficult to find in nature today and are no longer used include: *Veratrum* sp., *Rhamnus alpinum* ssp. *fallax*, *Gentiana lutea* and *Ribes uva-crispa.* The use of *Chenopodium album* has also died out. We can assume that the differences in ethnobotanical knowledge between the three studied areas are partly due to minor differences in climate and topography, while other causes lie in the higher degree of rurality and stronger ties to nature in the Lovinac and Perušić areas. The most important finding of the study is the use of the parasitic fungus *Taphrina pruni* (Fuckel) Tul. as a snack. The use of *Helleborus dumetorum* for ethnoveterinary practices is also worth noting. The traditional use of plants in the study area shows many signs of abandonment, and therefore efforts must be made to maintain the knowledge recorded in our study.

## 1. Introduction

Traditionalist societies seek to preserve their traditional and cultural values, aiming to transfer them to posterity. Ethnobotanical studies can help in this task [1,2,3]. In the last few decades, we have often heard that traditional food is healthier than today’s new, quick cuisine. Together with the growing importance of natural and organic foods, the use of alternative products is increasing [1]. Some wild food plants (WFPs) are bioactive functional foods that can help people maintain health and fight against a number of diseases [4,5,6]. Besides, WFPs are rooted in conventional food knowledge, which is an important component of local and autonomous food systems [7].

Most botanical studies conducted in Croatia are focused on exploring floristic composition, vegetation and threatened and endemic species [8]. To date, some ethnobotanical research has been carried out in Croatia, mainly in the coastal or peri-coastal areas of the country [9,10,11]. However, further from the sea and at higher altitudes, only the area around Knin [12] in the region of Northern Dalmatia has been documented. No ethnobotanical research has been conducted on this topic in the Lika area, but in 2007, “Natural healing with herbs from the Lika-Senj county habitats”, a concise book on the topic, was published by Kate Milković [13].

According to Łuczaj et al. [14,15,16], the region of Dalmatia is very interesting from the point of view of ethnobotany, given the combination of Slavic and Mediterranean influence affecting the number of wild plants used in everyday life. Generally, peasants from Slavic countries used to resort to just a few of the commonest wild greens, ignoring other species [17]. There are some exceptions such as regions inhabited by southern Slavs, i.e., the inhabitants of Herzegovina [18] and the coast of southern Croatia—Dalmatia [19], who seem to have used an exceptionally high number of wild leafy vegetables in nutrition, as pointed out by Moszyński [20].

The evolving dynamics of ethnobotanical knowledge transmission have been found to be affected by different drivers, including political [21], historical, geographical and cultural differences [22] between rural and urban [23] and historical ethnogenic circumstances. As far as we know, throughout history, different invaders crossed paths in the central Lika region, which was located on the border between the Ottoman Empire and Europe (16th–17th century). Since the Iron Age, nomadism in the transhumance livestock economy of Lika was conditioned by its natural constitution (mountainous area and harsh climatic conditions), which prevented other economic activities, such as agriculture, from emerging. Therefore, livestock was necessary for sustenance [24,25]. Locals had very close connections to the mountain ecosystem they lived in. Throughout the entire Middle Ages, the Dinaric Vlachs were nomadic pastoralists who lived without permanent settlements and were tied to their herds [24,25].

The western regions of the Dinarides were inhabited by the so-called White Vlachs, whose name is derived from the white woolen clothes they wore.

The related Black Vlachs lived in the eastern areas of the Dinarides and wore black woolen clothes. Croats differed from the Vlach population in never having engaged in nomadic cattle breeding but always having lived in permanent settlements. White Vlachs, who lived around Velebit and Dinara in the Middle Ages, were Roman Catholics, while the Black Vlachs were partly Bogumili, and partly members of the Orthodox church. This situation was disrupted by the Ottomans, who reached Lika in the 16th century. Until the beginning of the 18th century, there were no major settlements located where the town of Gospić is today. The settlers who arrived here were mostly craftsmen and farmers of diverse origin: Christianized Muslims, Bunjevci and Catholics from the coast and upper Pokuplje (Bosnia and Herzegovina). During the period of Turkish settlement, Perušić enjoyed the reputation of an advanced and rich district center, a fortress guarded by soldiers and surrounded by populous villages. At the beginning of the 18th century, the Bunjevci and the Vlachs were settled in Lovinac and engaged in animal husbandry [25,26].

The people of Lika have always been poor and humble, surviving on “proja” (a traditional flatbread based on cornmeal or a mixture of different flours and eggs), sheep’s milk, cheese, potatoes, beans, cabbage, onions and bacon. Due to the high birth rate in the 19th century, Lika became the province with the most visible degree of emigration in Croatia, both to richer regions and overseas [24]. 

Since the traditional consumption of useful plants in the central Lika region has never been documented, the general aim of this study is focused on edible food and feed, medicinal plants, and ethnoveterinary uses, but also includes plants used for religious or traditional ceremonies and tools. The aim was also to contribute to the knowledge of plant biodiversity in the area.

## 2. Results

During the surveys, 1082 valid use records were collected, in which the respondents identified 111 plant species from 50 plant families and five taxa of mushrooms and fungi belonging to five families (Table 1). As many as 80 species of plants were collected only from the wild, 27 species were cultivated, and four species were both wild and cultivated. 

The number of taxa per interview was between 15 and 49, with a mean of 27 taxa per interview (Median = 24, *SD* = ±8.3940). The most recorded taxa belong to Rosaceae (21) and Compositae (13) plant families. Some of the families were represented by only one or two species, but those species were frequently mentioned among respondents, such as the families Cornaceae (*Cornus mas* L., mentioned 33 times), Urticaceae (*Urtica dioica* L., 31 times) and Juglandaceae (*Juglans regia*, 30 times). 

The most frequently mentioned taxon in the survey is *Achillea millefolium* L. (Figure 1), with relative frequency RFC = 0.925. The next two most frequent species are *Rosa canina* (*RFC* = 0.900) and *Cornus mas* (*RFC* = 0.825). A total of 10 taxa have absolute frequencies of 30 or more, while 40 were mentioned at least 10 times.

Of the total of 111 plant taxa recorded, 47 were mentioned in all three survey localities, while 28 occurred in at least two localities. Some taxa were recorded only in one particular locality: Perušić—28 taxa, Lovinac—9 and Gospić—1. The similarity or correspondence (Jaccard Index) between the pairs of areas is *JI* = 57.14% for the pair Gospić–Lovinac, *JI* = 55.86% for the pair Lovinac –Perušić and *JI* = 54.37% for the pair Gospić–Perušić.

The most important plant species in relation to Smith’s S saliency indicator values are *Rosa canina* L. (*Sj* = 0.6725), *Achillea millefolium* (*Sj* = 0.6670), *Cornus mas* (*Sj* = 0.5627), *Sambucus nigra* L. (*Sj* = 0.5236) and *Urtica dioica* (*Sj* = 0.5050). 

However, looking at the values of the cultural value coefficient (*CV_e_*), we find that the three taxa with the highest cultural value are *Cornus mas* (*CVe* = 0.812), *Taraxacum* spp. (*CV_e_* = 0.380) and *Achillea millefolium* (*CV_e_* = 0.356) (Figure 2).

Respondents used plants most frequently as food or drink (55 taxa) and for medicinal purposes (46 taxa) (Table 2). They used 22 taxa exclusively as food, 11 taxa exclusively as medicine, and 21 for both food and medicinal purposes. On average, 13 taxa were recorded per interview for food or beverage (Median = 12, *SD* = ±4.682) and seven for medicinal purposes (Median = 6, *SD* = ±4.144). In addition, plants were used as livestock feed (33 species), to make alcoholic beverages (15 species), for ceremonial purposes (11 species, Figure 3) and to build or make useful objects (10 species).

The average number of use reports per plant is 11.24 for the whole study area. The taxon *Cornus mas* has the largest number of recorded uses (*UR* = 63), followed by *Prunus domestica* L. (*UR* = 49), *Juglans regia* (*UR* = 46), *Urtica dioica* (*UR* = 41) and *Achillea millefolium* (*UR* = 41) (Figure 4). 

*Cornus mas* is not only the most commonly used plant; together with *Tilia platyphyllos*, it also has the most different uses, namely five. However, for the latter plant, the total number of uses is only *UR* = 30. On average, the number of uses, i.e., use categories, is 1.72 per plant.

*Prunus domestica* has the highest UR in the food and drink category (*UR* = 31) and *Achillea millefolium* in the medicine category (*UR* = 37). In alcoholic beverages, *Juglans regia* L. (*UR* = 19) is used most for the production of sweet walnut liqueur. *Taraxacum* sect. Ruderalia is mostly used as livestock feed (*UR* = 21), and *Corylus avellana* L. is the most commonly used plant in the category of building and making useful items (*UR* = 11) (Table 3).

Measured with the informant consensus factor (*Fic*), respondents showed the highest level of agreement in the food and beverage (*Fic_all_* = 0.89) and ceremonial purpose (*Fic_all_* = 0.87) use categories (Figure 5). Fic values are above 0.5 for all use categories, indicating a relatively high level of consensus.

Regarding use as food or beverage, interviewees have the highest agreement regarding *Prunus domestica* (*FLs* = 96.88) and *Cornus mas* (*FLs* = 90.91) (Table 4). In the medicinal use category, *Achillea millefolium* has *FLs* = 100, which means that all respondents who named this plant classified it as a plant for medicinal purposes. Other frequently mentioned taxa have a *FLs* for medicine of less than 50. In the category of taxa for alcoholic beverages, the highest *FLs* is for *Juglans regia* (63.33), followed by *Prunus domestica* (*FLs* = 56.25) and *Cornus mas* (*FLs* = 51.52). In the animal feed use category, agreement is highest for the taxa *Taraxacum* spp. (*FLs* = 65.62) and *Urtica dioica* (*FLs* = 45.16). A large proportion of respondents indicated that the taxon *Corylus avellana* is, or has been, used in construction or for making useful items for the home or farm. Of the ten taxa with the highest *UR*, only *Cornus mas* is used for ceremonial purposes (*FLs* = 42.42).

The most commonly used plant parts are fruits (38% of all plant parts mentioned), leaves (23%), and flowers (21%) (Figure 6). Fruits are most commonly used as food, while the flower and leaf parts are most commonly used for medicinal purposes. 

A total of 284 uses of plants for medicinal purposes were recorded, most of which were for treating stomach aches (44), colds (25), and coughs (21). The greatest variety of plant species is used to treat diarrhea (6), open wounds (5) and coughs (5). The most frequently reported unique medicinal uses were the treatment of heart problems in humans (six use reports) and the treatment of erysipelas in pigs (four use reports). We found high fidelity levels regarding the use of certain plants for medicinal purposes with respect to treatment of stomach pains, colds, coughs, wounds, abdominal cramps and pain, diarrhea, earache, anemia and warts (Table 5).

Mushrooms or fungi were mentioned a total of 34 times by 21 of the 40 respondents (mean = 0.85). Therefore, on average, 1.6 taxa were mentioned per interview. *Agaricus* sp. (20 times) and *Boletus* sp. (6 times) were mentioned most frequently. Mushrooms are used as food, usually fried with eggs. A more recent method of preparation is in risottos or sauces (*Boletus* sp., *Cantharellus cibarius* Fr., and *Morchella* sp.). Three respondents mentioned eating the fruits of *Prunus domestica* infected with *Taphrina pruni* (Fuckel) Tul. fungus as children.

### Treatments and Disease Prevention in Animal Health—Lika

The results show the most frequently mentioned plant species used to treat various diseases and ailments in animals. In both people and animals (cows, pigs, chicken...), diarrhea is treated with dried fruits of *Rumex* spp. or a decoction of *Cydonia oblonga* L., *R. pulcher* L., and oak bark (*Quercus* sp.), which is sometimes even combined with *Hypericum perforatum* L. A decoction of St. John’s wort (*H. perforatum*) or juice squeezed from raw green bean leaves (*Phaseolus vulgaris* L.) is also used to wash wounds and scratches in cattle and to stop bleeding. Hellebore (*Helleborus dumetorum* Waldst. & Kit. ex Willd.) has long been used for cattle and horses to treat inflammation of the larynx or ears. The affected area is pierced with a cleaned hellebore root, which leads to irritation and the formation of an abscess that draws out the inflammation. Danewort (*Sambucus ebulus* L.) has been used to treat swine ersypelas in pigs, reduce swelling from snake bites in cows, and when extracting fever from the body.

*Silene vulgaris* (Moench) Garcke is used in religious ceremonies, brought to church for blessing, later salted and given to cattle to eat.

The largest number of taxa (101) and the highest mean number of taxa listed per respondent (27.05) were recorded in the locality of Perušić. Seventy-three taxa were recorded in the locality of Lovinac (25.31 per respondent) and 59 in Gospić (average 22.09). The average number of taxa per subject is statistically significantly higher in the Perušić locality than in the Gospić locality (ANOVA *F*(2,47) = 3.5, *p* = 0.038, Scheffe post hoc test *p* = 0.008). The average number of records per respondent is significantly higher in the Perušić locality than in the Gospić locality (ANOVA *F*_(2,47)_ = 3.5, *p* = 0.038, Scheffe post hoc test *p* = 0.008). One possible reason for this difference is the larger proportion of the population in the Perušić locality employed in activities related to agricultural and forestry resources. Most of the population of Gospić is not employed in the primary sector and lives in urban conditions.

## 3. Discussion

Considering the entire study area, 40 of the total 111 taxa recorded have absolute frequencies of 10 or more, and 10 have frequencies of 30 or more. The frequencies of taxa in individual areas range from zero to 15, and are highest in the Perušić locality (16 respondents). In the Gospić locality (11 respondents), there are four taxa with frequencies of 10 or more; in the Lovinac locality (13 respondents) there are eight; and in the Perušić locality there are 15. These data confirm the difference in ethnobotanical knowledge in the three studied areas could be due to the greater degree of urbanization around Gospić. The relative frequencies indicate a higher concentration around a lower number of known taxa around Gospić and Lovinac compared to Perušić. Namely, in the former two areas, three taxa are mentioned by all respondents (*RFC* = 1.00). In Perušić, there is no such instance for any taxon. The decrease in ethnobotanical knowledge of people moving to urban settings (like Gospić) can be expected and was even recorded in the Balkans from Kosovo [27].

The inhabitants of the Lika region use very few or almost no wild vegetables in their diet for the purpose of cooking (stews, soups, with meats). Their diet is based on crops that they can grow themselves in the garden, such as cabbage that can be lacto-fermented in the fall for the winter, potatoes, beans, garlic, onions, and corn for flour or polenta. This is in contrast with, e.g., the Zadar region of Dalmatia on the other side of the Velebit range, where a large number of wild vegetable taxa is used [15].

The five wild plant species with the highest frequency of mentions from the area of central Lika (this research) are also commonly used in the area of Knin [12]: *Rosa canina* 90% (Lika) vs. 65% (Knin), *Sambucus nigra* 82.5% (Lika) vs. 72.5% (Knin), *Cornus mas* 82.5% (Lika) vs. 65% (Knin), *Taraxacum* 80% (Lika) vs. 40% (Knin) and *Prunus spinosa* 82.5% (Lika) vs. 35% (Knin).

Comparing this research and the neighboring area of Knin [12], we noted 55 plant taxa used in both areas, but not always for the same purpose. Out of the high-frequency species in the area of central Lika, *Rubus caesius* (UR 24), *Carum carvi* (UR 21), and *Pyrus communis* (UR 8) were not recorded in Knin. Comparing the most frequently mentioned species with studies from other countries, we found some different uses, e.g., *Juglans regia* and *Mentha spicata* L. leaves with salt were used as traditional insect repellents in Bulgaria [28]. In addition, we found some similar or identical medicinal applications of the most commonly used species. For example, *Achillea millefolium* is used in tea to treat stomach pain in South Kosovo [29].

A large number of fruit species are still used to distil *rakija* spirits in the study area, including *Prunus domestica*, *Pyrus* spp., *Cydonia oblonga*, *Juniperus communis* and—nowadays quite rarely, due to longer collection time—*Cornus mas*. Plum brandy, so-called *lička šljivovica* (“Lika plum brandy”), is used most commonly.

The local cultivar of pears (*Pyrus communis* “Tepka”) is also commonly used in the area —it must be bletted to be tasty for consumption, and in the past, it was cut, strung on twine, dried and eaten on Christmas Eve. A dish called *oša* consists of a thick compote of dried pears or prunes/fresh plums with homemade bread. However, this tradition has been forgotten in the last 50 years. According to Hećimović-Seselja [30], fall pears are used to make pickles such as “turšija” (Turkish: Turs, Greek: Τουρσί-tours, Bulgarian: Tursia, Albanian: Turshiju). The traditional recipe has been preserved to this day and many families still use it.

The observed number of wild mushrooms used for food is relatively low for the habitat of Lika, which is very rich in fungi species due to the presence of mixed deciduous forests. Here only a few species of mushrooms are used for food, but harvesting and selling mushrooms (for purchase or privately along the highway) has been an important contribution to the household budget of the inhabitants of Lika for many years; they are very widespread, and about 15 kg can be harvested in one afternoon (oral statement of respondents). For comparison, in the Romanian Carpathians, 24 species are used by a similar sample of the local population of Ukrainians (on average 9.7 per respondent) [31]. A large number of mushroom taxa are also eaten in Poland, e.g., 76 in the Mazovia region— 9.5 per respondent [32]. On the other hand, similarly to Lika, few species of fungi are used on the coast of Croatia, e.g., 0.2 species of fungi were listed per interview on Krk and in Poljica [33]. Altogether, six species are used on Krk and four in Poljica. In this sense, Lika is similar to the above coastal populations. However, it differs in the low use of wild vegetables, similar to northern Slavic populations [17]. Although the more “herbophilic” attitude of the Slavic population (highlighted in previous studies) is not visible in this survey, the same statement applies to ethnobotanical research in South Kosovo [29]. An interesting feature of fungi use in the study area is the use of plum fruits deformed by the fungus *Taphrina pruni* (Fuckel) Tul. The food use of this fungus has never been recorded in ethnobiological literature. The situation is comparable to the use of corn smut *Ustilago maydis* (DC.) Cordacalled in the Mexican *huitlacoche* [34]—a gall formed on the stalks of maize, said to have been eaten fresh by local residents and with a slightly sour taste.

A large number of children’s snacks should be noted. Some of the respondents state that as children, apart from *Taphrina,* they used to nibble on young *Fagus sylvatica* leaves and young *Rosa canina* shoots. They also state that they sucked the flower of *Primula vulgaris* or juice from the stem of *Rumex acetosa*. In the municipality of Lovinac, there is also mention of sucking juice from the root of *Scorzonera villosa*. 

As shepherding has been the basis for the local economy for centuries, it must be noted that some knowledge on plants used to cure and feed animal persists in the area. In Lika, *Sambucus ebulus* has been used to treat swine erisypelas in pigs or to reduce swelling from snake bites in cows as well as to extract fever from the body. Similar uses were recorded in an earlier study by Hećimović-Seselja from 1985 [30]. In Poland *S. ebulus* was used to treat tuberculosis [35], and Akbulut & Özkan have recorded its use in Turkey against external parasites [36].

The majority of food and medicinal plants used in the area are ubiquitous species of widely known and documented use. *Helleborus dumetorum* Waldst. & Kit. ex Willd is an exception. Although *Helleborus* species have been known to be used in human and animal medicinal practices, they are highly toxic. Therefore, their use has been decreasing. Their application in veterinary practices in the area is an archaic feature. *Helleborus species* are still used in other parts of south-eastern Europe, but relatively rarely, e.g., in Transylviania *Helleborus purpurascens* Waldst. & Kit. is known for its use in immunotherapy, treatment of wounds, and as an antiemetic drug in ethnoveterinary medicine [37]. In the neighboring Bosnia and Herzegovina, *Helleborus odorus* Waldst. & Kit. ex Willd is used for medicinal purposes for humans, in the form of a decoction and infusion for liver and skin disease [38].

Another use of *Helleborus* is the children’s game “titra”, which uses the plant’s stems and is still remembered by the local population. Several players can participate in the game. According to the rules, each player can have 5 to 10 “titre”—stems of *Helleborus*. The stems are arranged in a bundle. The first player holds them in the palm of his right hand, transfers them to the upper side of the same hand, catches them and counts how many pairs of stems he has caught. If the player counts an odd number, then he takes one “titra”, puts it aside, and throws the rest of the stems one more time, palm facing down. If the player has an odd number of stems the second time, he sets two stems aside. If the player catches an even number during their next turn, they lose, and the next person follows the same procedure until all her credits are used up. The player with the most “titre” is the winner [30].

The effect of the ex-Yugoslav war between 1991–1995 may have also had a strong effect on plant knowledge in the area. The region was highly depopulated both by heavy fighting with the Serbian army and by the exodus of local inhabitants to cities and other countries. Additionally, a large percentage of the Orthodox (Serbian, Vlah) population left the region.

## 4. Materials and Methods

### 4.1. Study Site

The field research study was conducted in a continental, mountainous part of Croatia, in three municipalities (Perušić, Gospić and Lovinac) of Lika-Senj County (Figure 7). 

The areas of the municipalities extend from NW to SE and have a range of ca. 60 km, with an elevation of mainly 550–600 m above sea level. The study area covers 1692 km^2^ with a total of 16,390 inhabitants, most of whom live in Gospić (12,745 or 78%). Perušić and Lovinac are typical rural areas with a very low population density of 6.89 and 2.95 inhabitants per km^2^, respectively. The municipality of Gospić has the status of a town and is the seat of Lika County. Gospić is the only settlement with more than 10,000 inhabitants in the county, but even this administrative area is sparsely populated, with only 13.18 inhabitants per km^2^. The percentage of inhabitants aged 65 years or above is over 35% in the locality of Perušić and Lovinac, and 21% in the locality of Gospić.

According to its current population density, Lika is a sub-ecumenical area. From the beginning of the 20th century, Lika lagged far behind in terms of agricultural and economic development, and there was not enough food for the population, so some of the inhabitants moved to other parts of Croatia or abroad [26]. These periods coincided with periods of war, namely World War II and the war for Croatian independence (Homeland War, 1991–1995). According to Bušljeta Tonković [39], the processes of deagrarianization, deruralization, urbanization, aging and population decline, which are defined as socio-geographical factors of landscape development, are present in the area of central Lika. They have led to a series of negative features, such as the extensified use of agricultural areas and the abandonment of their cultivation, which have caused the landscape to show signs of neglect.

The cessation of animal husbandry and the abandonment of pastures and meadows has led to progressive succession, causing pastures to disappear and be replaced by scrubland and eventually forest, which is the climax (final stage) of vegetation in Croatia.

In the area of central Lika, the proportion of fertile land is small due to the domination of karstic rocks, which, along with unfavorable climatic conditions (low average annual temperature, short growing season for certain crops), significantly affects the characteristics of agricultural production (the choice of crops for cultivation is narrowed).

According to the Köppen climate classification, the area of Lika is dominated by a moderately warm humid climate with hot summers (Cfb), and only the highest mountain areas of Velebit, Plješivica and Mala Kapela have a snow-forest climate (Df) [40]. Accordingly, beech and fir forests (*Aremonio-Fagion*) predominate in Lika, and in the highest areas the beech forest transitions into low forms of coniferous trees, where *Pinus mugo* Turra is dominant. Downy oak and sessile oak (*Quercion pubescenti-petraeae*) predominate in fields, forests and thickets, where white hornbeam (*Carpinetum orientalis* s.l.) is also present [41].

The landscape of Lika includes characteristic slopes bounded by the longest Croatian mountain Velebit—the highest peak of Lika (Vaganski vrh, 1757 m)—in the west-southwest, by Velika Kapela and Mala Kapela in the northwest, and by Plješivica (highest peak Ozeblin, 1657 m) in the east [41].

The Velebit mountain range separates the continental and coastal parts of the Lika-Senj County, and the Northern Velebit National Park, the Paklenica National Park and the Velebit National Park lie on its territory [42]. Plitvička jezera NP, the oldest and most visited Croatian national park, is located in the northeastern part of the county and is included in the UNESCO list of world natural heritage. Between the highlands, at an altitude of 500 m above sea level, lie karst fields under agricultural use, the most important being Gacko polje and Ličko polje. The Gacka and Lika sinkhole rivers, which belong to the Adriatic basin, flow through these fields, and there is an artificial reservoir lake Kruščica, on the Lika river.

### 4.2. Sampling and Interviews

Data were collected through a survey of the local population conducted during the period February–May 2020. The second author of this paper is originally from the investigated area and helped with the selection of the interviewees. In-depth semi-structured interviews and the free listing method were used. The principles of the American Anthropological Association Code of Ethics [43] and the International Society of Ethnobiology Code of Ethics [44] were followed in conducting the survey. Respondents were selected using the snowball method [45]. Only native inhabitants or those who had lived there most of their lives were interviewed. Recommendations from key respondents were also considered in the respondent selection process. Interviewees were approached mostly outdoors, and interviews were conducted during walks so that informants could see, recognize, and name individual plants. During the survey, information was collected on the vernacular names of the plants, their useful parts, types of use, and preparation/recipies for use. Forty interviews were undertaken–16 in Perušić area, 11 in Gospić and 13 in Lovinac area. The mean age of respondents was 73 years, and the median age was 76 years. The average age of respondents from Lovinac area (67 years) is slightly lower than in the other two areas, but the age difference is not statistically significant (it ranged from 55 to 88). Females constituted the majority of informants (87.5%). The interviewees were of Croatian ethnicity from Roman-Catholic religious background.

Seven non-exclusive modalities or categories were available for various uses: food or drink, medicine, alcoholic beverages, animal feed, construction and handicrafts, ceremonial uses, and other. For plant identification, standard floras for this area of Europe were used, such as: Nikolić’s guide to the identification of the flora of Croatia [46], Pignatti’s Flora of Italy [47], and the Flora Croatica Database [48]. The names of the plants are aligned with The Plant List [49]. The voucher specimens were collected, digitized and stored at the University of Zagreb Faculty of Agriculture, ZAGR Herbarium (http://herbarium.agr.hr/ (accessed on: 14 October 2022)). Fungi were placed in the Croatian National Fungarium (CNF) in Zagreb (http://mycolsoc.hr/hrvatski-nacionalni-fungarij/ (accessed on: 14 October 2022)).

### 4.3. Data Analysis

The main variables of the dataset were: the informant’s code number, the place of the survey, the order of mention of the plants, the local plant name, the types of use and the preparation method for each type of use, i.e., the use category. In addition, for each taxon recorded, the following information was provided: scientific plant name and plant parts used.

All taxa mentioned two or more times were included in the analyses performed. Taxa that were mentioned only once were included only if we considered the information to be particularly relevant and provided by the most trusted informants.

To determine the level of awareness of a single taxon among respondents, relative frequencies were calculated using the following formula:*RFC* = *FC/N*,(1)
where *RFC* is the relative frequency of citation, *FC* is the absolute frequency or number of mentions of a single taxon, and *N* is the total number of informants [50].

The importance of a single taxon with respect to the entire study area and each of its three subregions was first evaluated using Smith’s S salience measure (*Sj*) [51]. This indicator is used to find taxa that are most characteristic of the studied area, taking the frequency and order of mention into account. The following formula was used to calculate *Sj*:*Sj* = ((*L* − *Rj* + 1)**/***L*)/*N*,(2)
where *L* is the length of the plant list per informant, *Rj* is the rank of taxon *j* in the list, and *N* is the total number of lists in the survey.

We assessed the practical importance of taxa to the community using the cultural value coefficient (*CV*) [52]. This indicator quantifies the usefulness of a taxon to the community based on the eight categories of use. The value of *CV* depends on the number of respondents who report using a taxon and the number of use categories. The following formula was used to calculate *CV*:*CV_e_* = *Uc_e_*·*Ic_e_*·*∑IUc_e_*,(3)
where *Uce* represents the number of uses of taxon e divided by the number of possible uses (in this case, 7). The symbol Ice represents the number of respondents who indicated that taxon *e* is used, relative to the total number of respondents, and *IUce* is the quotient of the sum of respondents for each of the 7 uses and the total number of respondents. For example, if taxon *e* was used for food by 35 respondents and for medicinal purposes by 4 respondents, *IUce* = (35 + 4)/40 [53].

The respondents’ consensus regarding the use of plants for a specific purpose was quantified using the informant consensus factor (*Fic*) function. This indicator is based on the number of use reports (*n_ur_*) and the number of taxa (*n_taxa_*) per specific use category [54,55]:*Fic* = (*n_ur_* − *n_taxa_*)/(*n_ur_* −1).(4)

The use report indicator (*UR*) per species is the number of mentions of use of a specific plant species in all use categories [55]. A higher value of this indicator suggests a higher level of agreement among respondents.

To determine the taxa that are most commonly used for a given purpose, we used the *FLs* function, which calculates the degree of fidelity per species. The *FLs* value for a given taxon and use category represents the percentage of informants using a plant for the same purpose compared to all uses of all plants:*FLs* = (*Ns* ∗ 100)/*FCs*.(5)

In the formula, *Ns* represents the number of informants using a given plant for a given purpose, and *FCs* represent the total number of uses for the taxon [56].

Calculations of descriptive statistics indicators for respondent characteristics, absolute and relative frequencies of plant species, the number of species per interview, and Jaccard similarity coefficients (JI) were performed using the program Microsoft Excel, Version 2013, Microsoft Corporation, Redmond, WA, USA. A comparison of results by location was performed using ANOVA in the program IBM SPSS Statistics for Windows, Version 21, IBM Corp., Armonk, NY, USA. Other calculations and analyzes were performed using packages “ethnobotanyR” [55] and “AnthroTools” [57] of the R, Version 4.0.3, R Foundation for Statistical Computing, Vienna, Austria [58].

## 5. Conclusions

The list of plant taxa used is largely typical for central Europe. Like in other central European regions, few wild vegetables are used, and mainly fruits and medicinal herbs are collected from the wild. Similarly to coastal parts of Croatia, surprisingly few taxa of edible fungi are collected, in spite of the presence of large areas of woodland.

We can assume that the differences in ethnobotanical knowledge between the three studied areas are partly due to minor differences in climate and topography, while other causes lie in the higher degree of rurality and stronger ties to nature in the Perušić and Lovinac areas.

Based on the observed situation and results, we propose actions to preserve the remaining traditional plant knowledge, primarily in the form of activities and workshops to promote the use and preparation of local wild plant food products and herbal teas among the local population. 

## Figures and Tables

**Figure 1 plants-11-03133-f001:**
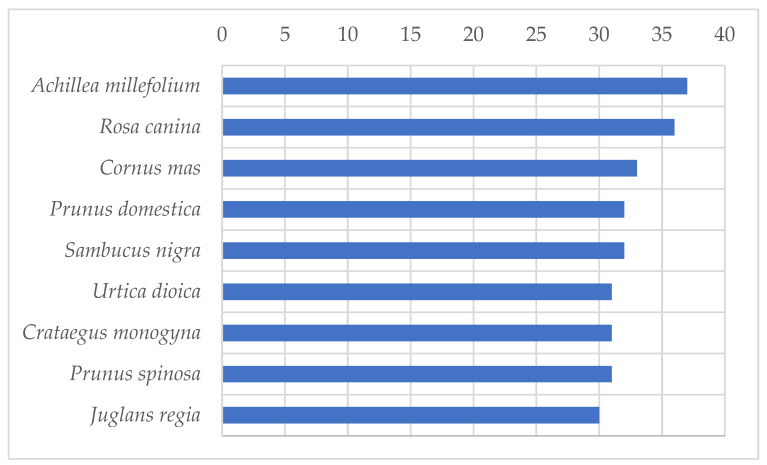
Plants with the highest absolute frequencies (Frequencies > 30), N = 40.

**Figure 2 plants-11-03133-f002:**
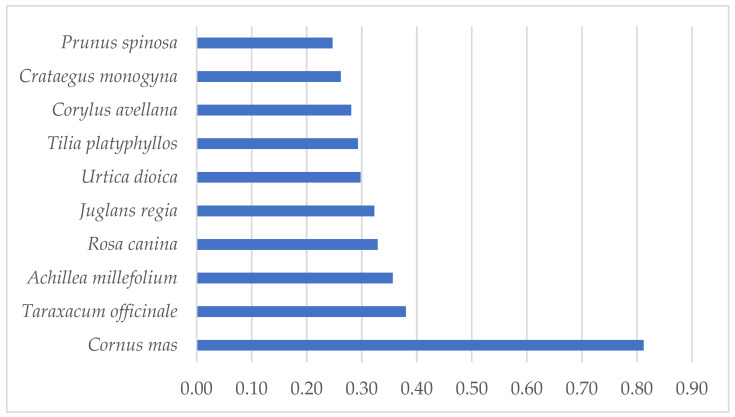
Ten species with the highest cultural value coefficients.

**Figure 3 plants-11-03133-f003:**
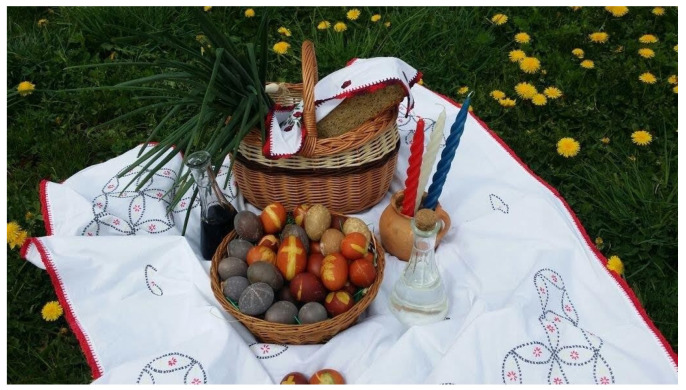
Baskets with food to be taken to church for blessing on Holy Saturday, including traditional Easter eggs colored with onion skins (photo: Hodak A.).

**Figure 4 plants-11-03133-f004:**
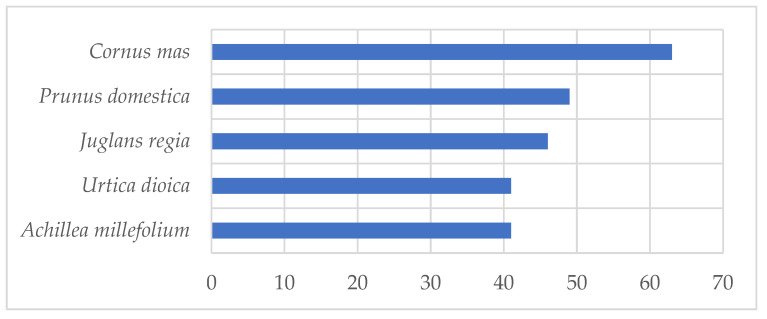
The first 15 highest ranked taxa according to use reports (*UR*).

**Figure 5 plants-11-03133-f005:**
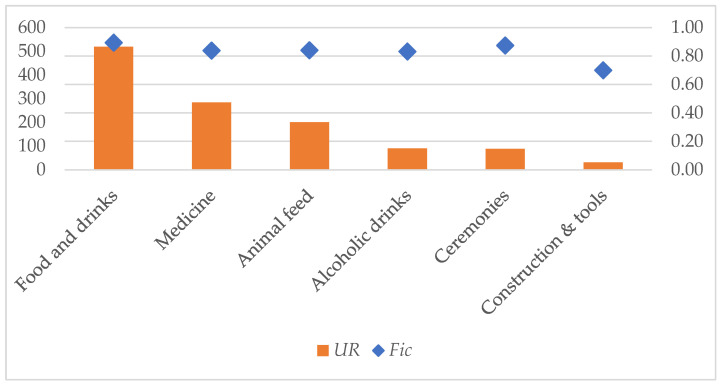
Total use reports and factor of informant consensus (Fic) for specified use categories.

**Figure 6 plants-11-03133-f006:**
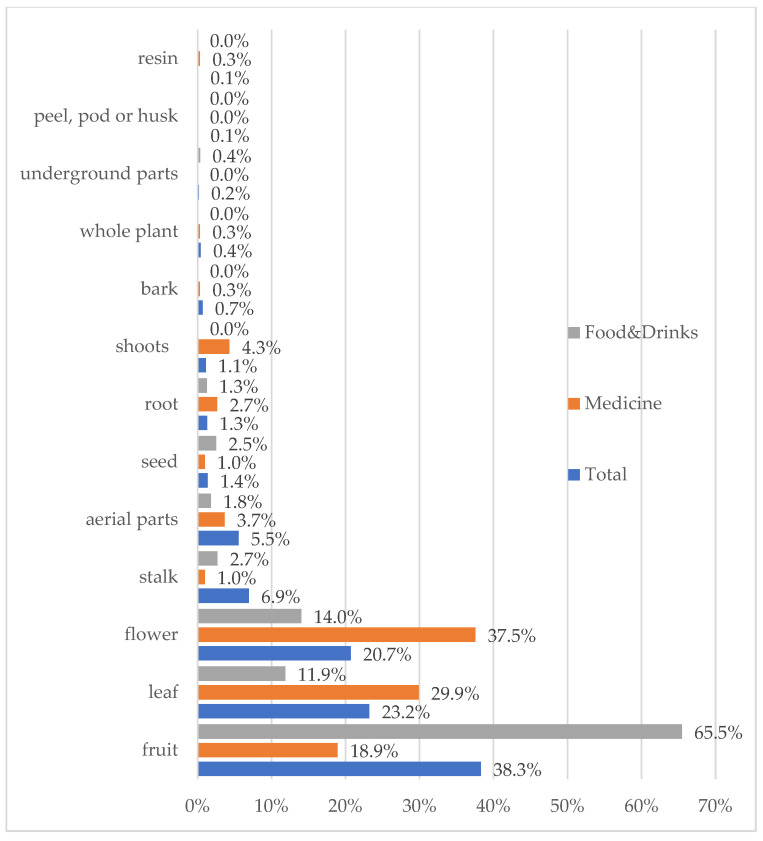
Relative frequency of mentioning plant parts.

**Figure 7 plants-11-03133-f007:**
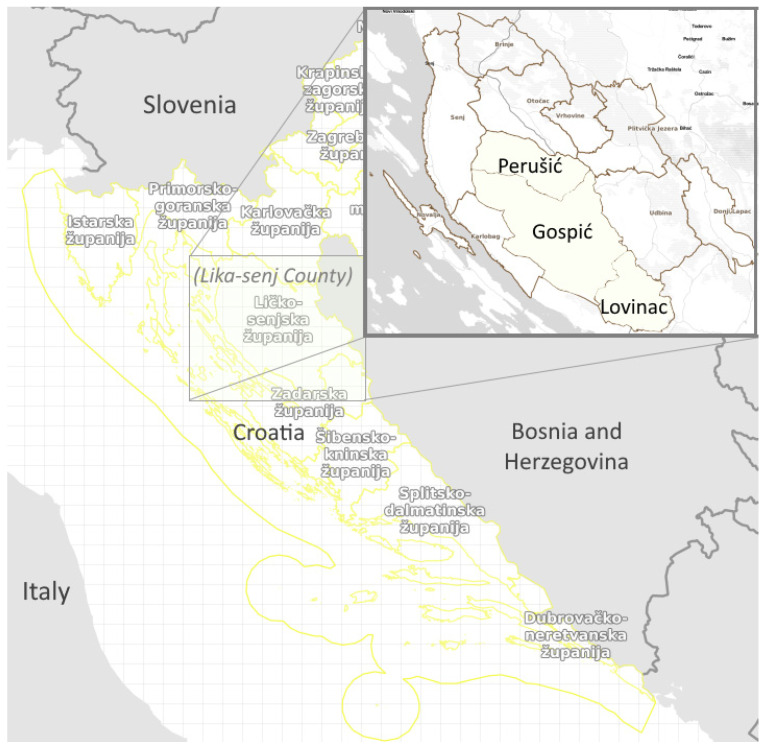
The geographical position of the central Lika study site (source: based on digital maps of Lika-Senj County, Geoportal [https://www.rc.licko-senjska.hr/geoportal/ (accessed on: 3 November 2022)] and Informacijski sustav prostornog uređenja, Geoportal, [https://ispu.mgipu.hr/ (accessed on 3 November 2022)]).

**Table 1 plants-11-03133-t001:** List of documented wild and cultivated plant taxa used in central Lika region (continental Croatia).

Botanical Taxon and ZAGR ID	Family	FC (N = 40)	Use Categories	Status	Local Name	Used Part	Use
*Abies alba* Mill. (ZAGR61711)	Pinaceae	11	MD, CT, CE	w	borove iglice, četina, jela	wh	honey-like syrup made from young needles, furniture making (so-called “tripod” chair), tambourine making
*Acer obtusatum* Waldst. & Kit. ex Willd. (ZAGR56773)	Sapindaceae	2	CT	w	javor	st	tambourine making
*Achillea millefolium* L. (ZAGR56740)	Compositae	37	MD, AL, AF	w	ajdučica, armanac, hajdučica, raman, sporiž, stolisnik, šporiž	fl	tea for the stomach, tea against colds, leaves mixed with cornmeal as additional food for turkeys
*Aesculus hippocastaneum* L. (ZAGR56503)	Sapindaceae	2	MD	w	divlji kesten	fr	tincture (brandy and horse chestnut) for poor circulation and rheumatic complaints, used externally
*Agrimonia eupatoria* L. (ZAGR36271)	Rosaceae	2	MD	w	kuročičak	lf	tea for the treatment of the liver, diarrhea, and poor bile secretion
*Allium ampeloprasum* L. (ZAGR56838)	Amaryllidaceae	3	FO	w	divlja kapula, divlji luk, pasiluk	fr, ap	used in the diet, raw, in salads; in the past, snacked by children directly from the meadow
*Allium cepa* L.	Amaryllidaceae	2	MD, CE	c	kapula	lf	cuticle stops cuts from bleeding; the scaly outer leaves of the bulb are used to dye Easter eggs
*Amaranthus retroflexus* L. (ZAGR39955)	Amaranthaceae	2	AF	inv	šćir, štir	st, lf	pig fodder
*Amelanchier ovalis* Medik.	Rosaceae	2	FO	w	merala	fr	fruits eaten raw, sometimes processed into marmalades or juices
*Arctium lappa* L. (ZAGR56771)	Compositae	4	AF	w	repušina	lf	pig fodder
*Artemisia absinthium* L. (ZAGR56747)	Compositae	17	MD, AL	w	pelim, pelin	lf, st	tea for the stomach, put in brandies for the stomach
*Arum maculatum* L. (ZAGR56491)	Araceae	16	AF	w	kozlac, volak	lf	pig fodder
*Avena sativa* L. (ZAGR56728)	Poaceae	2	AF	c	zob	fr	horse fodder
*Bellis perenis* L. (ZAGR56487)	Compositae	3	CE, OT	w	tratinčica,	fl	steeped in cold water for ceremonial face-washing with flowers for Palm Sunday
*Beta vulgaris* L. spp. crassa	Amaranthaceae	3	AF	c	kravlja ripa, ripa	fr	cow and pig fodder
*Betula pendula* Roth (ZAGR56729)	Betulaceae	2	FO, MD	w	breza	lf	tea for diabetics; in past, young leaves were eaten by children as snack
*Brassica oleracea* L.	Brassicaceae	2	FO, MD	c	kupus	lf	medicine (raw leaf is used externally for sprains)
*Calendula arvensis* M.Bieb.	Compositae	2	MD	w/c	neven	fl	ointment for treatment of lichen on the skin
*Cannabis sativa* L.	Cannabaceae	7	OT	c	konoplja	ap, st	spun for weaving straw mats (mattresses, straw mattress pads), folk costumes, bed linen
*Cirsium arvense* (L.) Scop. (ZAGR56833, 56834, 56835)	Compositae	15	AF	w	boda, bodež, oslobad, sikavac	ap, lf	pig fodder
*Carum carvi* L. (ZAGR43948)	Apiaceae	22	FO, MD	w/c	kimljan, kimljen	fr, sd	tea against stomach pain
*Chelidonium majus* L. (ZAGR56475)	Papaveraceae	3	MD	w	rosopas	ap, st	treatment of skin warts
*Chenopodium album* L.	Amaranthaceae	13	AF	w	loboda	lf	pig fodder
*Cichorium intybus* L. (ZAGR56802)	Compositae	4	FO	w	konjogriz	rt, fl, lf	tea against diarrhea
*Convolvulus arvensis* L. (ZAGR56777)	Convolvulaceae	14	AF	w	slak, slatkovina	ap	pig fodder
*Cornus mas* L. (ZAGR 56480, 56476)	Cornaceae	33	FO, MD, AL, CT, CE	w	dren, drenila, drenjine, drin, drinak, drin’ć, drinić, drinle, drin’le, drinjine	fr, ap, fl,	juice; jam; liqueur; brandy; wine; handle for cooks; twig blessed on Palm Sunday
*Corylus avellana* L. (ZAGR 56481)	Betulaceae	25	FO, AF, CT, OT	w	lijeska, liska, lišnjak, ljeska	fr, ap, st, fl	for food, flowers (aglet) are cooked with flour; food for pigs; stakes for the garden; fence weaving
*Crataegus monogyna* Jacq. (ZAGR56743)	Rosaceae	31	MD, AL	w	glog, glogić, gloginja, trnina, trnkulica	fr, fl, lf	in brandy; tea as a medicine against atherosclerosis, hypertension, to strengthen heart function and to soothe, for nervousness
*Crocus* sp. (ZAGR56489)	Iridaceae	4	OT	w	brnduša, šafran	lf, fl	collected by children for a game
*Cydonia oblonga* Mill. (ZAGR56767)	Rosaceae	20	FO, MD, AL	c	dunja	lf, fr	tea from fruit, tea from leaf (against diarrhea), jam, juice, brandy
*Elymus repens* (L.) Gould (ZAGR39917)	Poaceae	2	AF	w	pirika	ae	pig fodder
*Fagus sylvatica* L. (ZAGR56809)	Fagaceae	14	FO, AF, CT, OT	w	bukva	lf, st	young leaves were eaten by children in the past; making household items (cutting board, bread board), furniture (“tronožac“ chair—A small wooden chair with three legs), tambourine making; feed for sheep and cattle
*Fragaria vesca* L. (ZAGR56732, 56495)	Rosaceae	23	FO, CE, OT	w	divlja jagoda, šumska jagoda	fr, fl	raw fruit
*Fraxinus excelsior* L. (ZAGR56502)	Oleaceae	7	MD, AF, OT	w	jasen	bk, lf	trumpet making; food for animals
*Galanthus nivalis* L. (ZAGR56484)	Amaryllidaceae	2	OT	w	visibaba	fl	cut flowers as decoration
*Gentiana lutea* L. ssp.symphyandra (Murb.) Hayek	Gentianaceae	2	MD	w	srčenik, srčanik	rt	medicine for the stomach (a piece of the root is soaked in brandy)
*Hedera helix* L. (ZAGR56829)	Araliaceae	13	CE	w	bršljan	ap, lf, fl	mixed with *Vinca, Primula* and *Viola* as wedding decoration; Flower Sunday bouquet,
*Helleborus dumetorum* Waldst. & Kit. ex Willd. (ZAGR56844)	Ranunculaceae	9	MD, OT	w	sitnocvjetni kukurijek, kukurijek, titra, kukurik	ap, st, rt	used in a children’s game (“titre”); medicine for cows and oxen
*Hordeum vulgare* L.	Poaceae	2	FO, AF	c	ječam	sd	feed for animals; substitute for coffee
*Hypericum perforatum* L. (ZAGR56500, 56811)	Hypericaceae	12	FO, MD	w	cvit svetog Ivana, gospina trava, kantarion	fl, lf	calming tea; tea for washing wounds in animals and humans
*Juglans regia* L. (ZAGR56742,56801)	Juglandaceae	30	FO, AL, CT	c	orah	fr, st	preparation of various dishes and cakes, liquor; making weapons (rifle butts, knife holders); making furniture
*Juniperus communis* L.	Cupressaceae	7	FO, MD, AL	w	borovica	fr	brandy for massage against rheumatism, tea, spice
*Linum usitatissimum* L. (ZAGR56730)	Linaceae	2	CT	c	lan	st	spun for linen fabrics
*Lotus corniculatus* L. (ZAGR56776)	Leguminosae	2	AF	w	divlja smiljkita	ap	animal fodder
*Malus domestica* Borkh. (ZAGR56474)	Rosaceae	19	FO, MD, AL, AF	c/w	divlja jabuka, divljakinja, divljankinja	fr	tea, animal feed, vinegar
*Matricaria chamomilla* L (ZAGR56505)	Compositae	14	FO, MD	w	kamilica	fl	tea against stomach pain
*Melissa officinalis* L.	Lamiaceae	2	FO	w	matičnjak	lf	calming tea
*Mentha* spp. (ZAGR56841,56840,56842)	Lamiaceae	9	FO, MD	w	divlja metvica, metvica	lf	tea to calm the stomach
*Mespilus germanica* L.	Rosaceae	2	FO	c	mušmula	fr	food preparation; fruit consumed raw
*Morus alba* L. (ZAGR56499)	Moraceae	10	FO	c	dud, murva	fr	food preparation; fruit consumed raw
*Nicotiana tabacum* L.	Solanaceae	2	OT	c	duhan	lf	in the past grown for smoking
*Panicum miliaceum* L.	Poaceae	2	FO	c	proso	sd, fr	past use for porridge
*Phaseolus vulgaris* L.	Leguminosae	4	FO, MD, AF	c	grah, komunje, grahorina	lg, lf, fr	juice squeezed from raw green bean leaves for washing wounds in animals;fruit of the bean was placed on the cut to stop the bleeding
*Picea abies* (L.) H.Karst.	Pinaceae	7	MD, CE	w	bor, četina, smreka	sh, nd, rs	“honey” from young pine needles, resin for wound healing (extracts pus)
*Plantago lanceolata* L. (ZAGR56477)	Plantaginaceae	11	MD	w	bokvica, trputac uskolisni	lf	honey against cough, resin against wound suppuration
*Plantago major* L. (ZAGR56770)	Plantaginaceae	8	MD	w	bokvica, trputac širokolisni	lf	lard and leaf used as healing agent for treating wounds
*Primula vulgaris* Huds. (ZAGR56485)	Primulaceae	17	FO, CE, OT	w	jaglac, pivčić	fl	traditionally used by children to wash themselves for Palm Sunday; flowers eaten; making musical instruments
*Prunus avium* (L.) L. (ZAGR56727)	Rosaceae	7	FO	c	divlja trišnja, šlama, trišnja	fr	used for food preparation; raw fruit eaten
*Prunus domestica* L. (ZAGR56768)	Rosaceae	32	FO, AL	c	šljiva	fr	brandy, jam, juice, liqueur, raw fruit eaten
*Prunus domestica* subsp. *insititia* (L.) Bonnier & Layens	Rosaceae	4	FO	c	bela šljiva, cibura, cimbura	fr	food preparation, jam
*Prunus mahaleb* L. (ZAGR38902)	Rosaceae	2	FO	w	rašeljka	fr	raw fruit
*Prunus spinosa* L. (ZAGR56739, 56805)	Rosaceae	31	FO, MD, AL	w	crni trn, glog, gloguljica, šljivov glog, trnila, trnina, trn’le, trnjina	fr	eaten when soft (after frost); tea; liqueur
*Pulmonaria officinalis* L. (ZAGR56492)	Boraginaceae	2	FO	w	plućnjak	fl	children sucked the juice from the flower
*Pyrus communis* L. (ZAGR56775)	Rosaceae	14	FO, AL	c	kruška, jesenka, kruškica	fr	dried fruit compote, pickles, brandy, eaten after bletting
*Pyrus communis* L. ‘Tepka’ (ZAGR56775)	Rosaceae	6	FO, AL	c	jesenka	fr	traditionally processed into juice in combination with apple; whole fruits are dried in a bread oven to make compote or ground and added to bread flour; most often used to make fruit brandy
*Pyrus pyraster* (L.) Burgsd. (ZAGR56807)	Rosaceae	8	FO	w	divlja kruška, divlja kruškica	fr	tea
*Quercus petraea* (Matt.) Liebl.	Fagaceae	2	AF	w	žir	fr, lf, bk	pig fodder
*Quercus robur* L. (ZAGR56498)	Fagaceae	16	MD, AF	w	hrast, rast, žir, žirevina	fr, lf, bk	pig, sheep and cattle fodder
*Rhamnus alpina* subsp. *fallax* (Boiss.) Maire & Petitm.	Rhamnaceae	2	OT	w	žestika	bk	used to be collected in the area and sold for medical purposes
*Ribes uva-crispa* L.	Grossulariaceae	5	FO	c	ringuze, runguza, ogrozd, šmanjak	fr	human food
*Robinia pseudoacacia* L. (ZAGR56501, 56810)	Leguminosae	2	FO	inv	akacija	fl	juice, tea
*Rosa canina* L. (ZAG56504)	Rosaceae	36	FO, MD, AL	w	šipak	fr	jam, tea, children used the young shoot as a snack
*Rubus caesius* L. (ZAGR56734)	Rosaceae	24	FO, MD	w/c	kupina	fr, lf	raw food, tea against diarrhea
*Rubus idaeus* L.	Rosaceae	14	FO, AL	c	malina	fr	liqueur, raw food
*Rumex acetosa* L. (ZAGR56496)	Polygonaceae	10	FO	w	kisela trava, kiselica, ljutak, ljutika	st, ap, lf	juice from green parts squeezed out and drunk by children
*Rumex pulcher* L. (ZAGR56804, 56769)	Polygonaceae	23	MD, AF	w	konjski štap, štavalj, štavelj, štavlina	fr, lf, sd	animal; medicine against diarrhea for humans and animals
*Salix purpurea* L. (ZAGR56493)	Salicaceae	4	CT, OT	w	rakita	st	weaving baskets; making instruments (trumpets)
*Salvia officinalis* L.	Lamiaceae	7	FO, MD	w/c	kadulja	lf	tea for sore throat (gargle)
*Sambucus ebulus* L. (ZAGR56774)	Adoxaceae	7	MD, AF	w	aptovina	lf	against Swine Erysipelas; poultice for cows in case of snake bites
*Sambucus nigra* L. (ZAGR56746)	Adoxaceae	32	FO, MD	w	bazga, zobka, zobika	fl, fr	tea, syrup
*Satureja montana* L.	Lamiaceae	2	FO, MD	w	vrisak	st, lf	honey, spice
*Satureja subspicata* Bartl. ex Vis. (ZAGR56731)	Lamiaceae	5	FO, MD	w	planinski vrisak, vrisak	fl	tea against cough
*Scorzonera villosa* Scop. (ZAGR56744)	Compositae	3	FO, AF	w	turutva	rt, fl	children sucked the flower, food for sheep (they love to eat it)
*Secale cereale* L. (ZAGR56737)	Poaceae	4	FO, CT	c	raž	sd, st	flour; straw as roof thatching
*Sempervivum tectorum* L.	Crassulaceae	11	MD, OT	w	čuvarkuća	ap	against earache; decoration on old houses
*Silene vulgaris* (Moench) Garcke (ZAGR 56830, 56831)	Liliaceae	4	CE	c	cvijet sv. Ante, trava sv. Ante	ap, st, fl	decoration; tea for various ailments; blessed, salted and given to cattle during the feast of St. Antonius
*Silybum marianum* (L.) Gaertn.	Compositae	3	MD, AF	w	sikavac	fr, lf	tea for treatment of the liver and gall bladder
*Sorbus aria* (L.) Crantz (ZAGR56748)	Rosaceae	9	FO	w	jarebika, mukinja, mukulja	fr	raw snack
*Sorbus domestica* L. (ZAGR56726)	Rosaceae	6	FO	w	oskoruša	fr	raw snack
*Sorbus torminalis* (L.) Crantz (ZAGR56799,56745)	Rosaceae	12	FO	w	brekinja, brekulja	fr	raw snack
*Symphytum officinale* L. (ZAGR56803)	Boraginaceae	5	MD	w	gavez	rt, lf	ointment against rheumatism (based on lard)
*Tanacetum balsamita* L. (ZAGR56808)	Compositae	4	CO	c	galoper, kaloper	lf	fragrance for women
*Taraxacum* sect. Ruderalia (ZAGR56497)	Compositae	32	FO, MD, AF, OT	w	maslačak, mličak, radič, radić	lf, fl, st	“honey” (syrup with sugar) against cough; leaf used as food for pigs
*Thymus serpyllum* L. s.l. (ZAGR56478)	Lamiaceae	13	FO, MD	w	majčina dušica	lf, fl, st	tea against cold, spice
*Tilia platyphyllos* Scop. (ZAGR56733)	Malvaceae	25	FO, MD, AL, CT, OT	w	lipa	fl, lf, st	calming tea, tambourine making, cutting board, bread kneading bord
*Trifolium pratense* L. (ZAGR56472)	Leguminosae	9	AF	w	ditelina, divla ditelina, divlja ditelina, djetelina	ap, st, lf	pig fodder
*Trifolium repens* L.	Leguminosae	3	AF	w	ditelina, djetelina	ap	cattle and lamb fodder
*Triticum aestivum* L. (ZAGR56735)	Poaceae	2	FO	c	pir, pšenica	sd	flour
*Triticum spelta* L.	Poaceae	2	FO	c	pir	sd, fr	flour
*Tussilago farfara* L.	Compositae	3	AF	w	podbjel, repušina	lf	pig fodder
*Ulmus glabra* Huds. (ZAGR56736)	Ulmaceae	2	AF	w	brest, brist	lf	pig fodder
*Urtica dioica* L. (ZAGR56494)	Urticaceae	31	FO, MD, AF	w	kopriva	lf, fl	tea, juice, supplementary feed for poultry (with eggs and cornmeal)
*Valerianella locusta* (L.) Laterr. (ZAGR56482)	Caprifoliaceae	2	FO	w	divlji matovilac, matovilac	st	salad
*Veratrum* sp. (ZAGR56832)	Melanthiaceae	4	MD	w	čemerika	lf, fr	purchased for medical purposes (PLIVA)
*Verbascum pulverulentum* Vill. (ZAGR56741, 56772)	Scrophulariaceae	3	MD, AF	w	divizma	st, lf	tea against colds; animal fodder (cooked leaves)
*Verbena officinalis* L.	Verbenaceae	2	MD	w	sporiž	ap	tea against diarrhea
*Vinca minor* L. (ZAGR56479)	Apocynaceae	11	CE	w	pavenka	lf, st	wedding bouquets; decoration of the yard; rose-shaped decoration for wedding parties (lapel); used to be placed on the top of the flag (banner)
*Viola odorata* L. (ZAGR56488)	Violaceae	27	CE, OT	w	fijolice, ljubica, ljubičica	fl	steeped in cold water for washing children’s faces for Palm Sunday, alone or mixed with primroses (Primula vulgaris)
*Viscum album* L.	Santalaceae	6	MD, AF	w	imela, lišaj, mela	wh	pig fodder
*Vitis vinifera* (*Vitis labrusca* x *Vitis riparia)*	Vitaceae	2	AF	c	tudum (grožđe)	fr	brandy
*Zea mays* L.	Poaceae	3	FO, AF, OT	c	kukuruz	sd, fr	flour; animal feed; straw for compost or making straw mattresses
**Fungi**							
*Agaricus campestris* L. s. l. (CNF 1/8898)	Agaricaceae	20	FO	w	grljak (LO; GS), pečurka (PE)	fruiting body	fried or fried with eggs
*Boletus* section *Boletus,* including *B. aereus* Bull. ex Fr. (CNF 1/8897)	Boletaceae	6	FO	w	vrganj	fruiting body	fried with eggs, recently in risotto
*Cantharellus cibarius* Fr. s.l. (CNF 1/8899)	Cantharellaceae	4	FO	w	lisičarka	fruiting body	fried with egg, in sauce or risotto
*Morchella* sp.	Morchellaceae	1	OT	w	smrčak	fruiting body	recent use, for sauces
*Taphrina pruni* (Fuckel) Tul.	Taphrinaceae	3	FO	w	rogač na šljivi	deformed plums attacked by the fungus	eaten by children in the past

Abbreviations: types of use: MD—medicinal, CT—construction and tools, CE—ceremonial use, AF—animal feed, AL—alcohol, FO—Food and drinks, OT—other unspecified ways of use. Status: w—wild, c—cultivated, INV—invasive. Part used: ap—aerial parts, bd—buds, bk—bark, bl—bulb, cs—cell sup, fj—fruit juice, fl—flowers, fr—fruit, im—immature fruits, lf—leaves, lg—legumes, nd—needles, p—peel, po—pollen, pe—petals, pcl—pedicels, pc—pericarp, rs—resin, rt—roots, sd—seeds, sh—shoots, st—stalk, sty—stylus, tb—tuber, trt—taproots, u—underground parts, wh—whole plant.

**Table 2 plants-11-03133-t002:** Use reports and the number of plants in different use categories.

Use Category	Use Reports	Number of Taxa	Taxa Per Informant			
			Median	Mean	St. dev.	*C.V.*
Food and drinks	524	57	12	13.1	±4.8	37%
Medicine	285	48	6	7.1	±4.1	58%
Alcoholic drinks	90	16	2	2.2	±1.1	51%
Animal feed	201	33	4	5.0	±2.9	57%
Construction or hand craft	31	10	0	0.8	±1.4	176%
Ceremonies	85	11	2	2.1	±1.5	69%
Other not specified	43	18	1	1.1	±1.4	130%

**Table 3 plants-11-03133-t003:** Comparative overview of taxa according to *UR* in six use categories.

Food and Drinks		Medicine	
Taxa	UR	Taxa	UR
*Prunus domestica* L.	31	*Achillea millefolium* L.	37
*Rosa canina L*.	30	*Carum carvi* L.	21
*Cornus mas* L.	30	*Tilia platyphyllos* Scop.	14
*Prunus spinosa* L.	28	*Rumex pulcher* L.	14
*Juglans regia* L.	25	*Taraxacum* sect. Ruderalia	13
**Alcoholic Drinks**		**Animal Feed**	
Taxa	UR	Taxa	UR
*Juglans regia* L.	19	*Taraxacum* sect. Ruderalia	21
*Prunus domestica* L.	18	*Rumex pulcher* L.	18
*Cornus mas* L.	17	*Quercus robur* L.	16
*Pyrus communis* L.	8	*Arum maculatum* L.	16
*Artemisia absinthium* L.	8	*Cirsium arvense* (L.) Scop.	15
**Construction or Hand Craft**		**Ceremonial Purposes**	
Taxa	UR	Taxa	UR
*Corylus avellana* L.	11	*Viola odorata* L.	27
*Secale cereale* L.	4	*Cornus mas* L.	14
*Salix purpurea* L.	3	*Hedera helix* L.	13
*Fagus sylvatica* L.	3	*Vinca minor* L.	11
*Juglans regia* L.	2	*Primula vulgaris* Huds.	8

**Table 4 plants-11-03133-t004:** Fidelity Level (FLs) for the ten taxa with the highest UR.

	URs, Total	Food and Drinks	Medicine	Alcoholic Drinks	Animal Feed	Construction and Hand Craft	Ceremonial Purposes
*Cornus mas*	63	90.9	3.0	51.5		3.0	42.4
*Prunus domestica*	49	96.9		56.2			
*Juglans regia*	46	83.3		63.3		6.7	
*Achillea millefolium*	41		100.0	2.7	8.1		
*Urtica dioica*	41	74.2	12.9		45.2		
*Rosa canina*	39	83.3	22.2	2.8			
*Taraxacum* spp.	38	9.4	40.6		65.6		
*Corylus avellana*	36	76.0			20.0	44.0	
*Crataegus monogyna*	36	77.4	35.5	3.2			
*Sambucus nigra*	35	75.0	34.4				

**Table 5 plants-11-03133-t005:** Priority taxa for the treatment of certain health problems based on the fidelity level index.

Health Problem	Taxa	Fidelity Level
Stomach pains	*Gentiana lutea* L. ssp. *symphyandra* (Murh) Hayek, *Mentha* sp.	1.000
Cold	*Sambucus nigra* L., *Verbascum pulverulentum* Vill.	1.000
Cough	*Thymus serpyllum* L.	1.000
Wounds	*Prunus spinosa* L.	1.000
Abdominal cramps and pain	*Matricaria chamomilla* L.	1.000
Diarrhoea	*Cornus mas* L., *Cydonia oblonga* Mill., *Satureja subspicata* Bartl. ex Vis.	1.000
Earache	*Sempervivum tectorum* L., *Viscum album* L.	1.00
Anemia	*Urtica dioica* L.	1.00
Warts	*Chelidonium majus* L.	1.00
Tension (for relaxation)	*Tilia platyphyllos* Scop.	0.93
Diarrhea in cattle	*Quercus robur* L.	0.50
Hart issues	*Crataegus monogyna* Jacq.	0.60
Erysipelas in pigs (“Vrbanac”)	*Sambucus ebulus* L.	0.57
Painful joints	*Symphytum officinale* L.	0.60
Diabetes	*Phaseolus vulgaris* L.	0.50
Purulent inflammation	*Plantago major* L.	0.22

## Data Availability

The data presented in this study are available on request from the corresponding author.

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
