# Peer review of "Traditional Ethnobotanical Knowledge of the Central Lika Region (Continental Croatia)—First Record of Edible Use of Fungus Taphrina pruni"

_plants, 2022, doi:10.3390/plants11223133_

Round 1

Reviewer 1 Report

This article presented the use of wild and some cultivated plants, mushrooms and fungi in Lika region. The most important finding of the study is the use of the parasitic fungus Taphrina pruni (Fuckel) Tul. as a snack. The study will be useful for ethnomedicinal uses of flora. Before recommending this article for publication, there are some shortcomings for that should be resolve.

Abstract line 13 what is difference between mushrooms and fungi.

Methods should me briefly mentioned in the abstract.

Also add conclusion and future recommendation of the study in the abstract.

Line 19-20 the sentence must be revised not clear grammatically.

Line 25 change areas into area.

Line 29-31 the sentences are discussion which should not be add in the abstract.

Line 42 should be cited with relevant study.
DOI: 10.56042/ijtk.v21i3.31454

Line 45 must be cited with other relevant studies. The following studies could be helpful. https://doi.org/10.1016/j.chnaes.2021.08.002

In the introduction section line 85-89 should be cited with relevant study.

There is no need of Figure 2

Add future recommendations in the conclusion.

 The author should revised language and grammar of the study as well.

Author Response

Dear Reviewers

we corrected the text to nearly all of your comments (see the answers below)

Revision 1

Abstract line 13 what is difference between mushrooms and fungi.

>mushrooms were redundant, we removed it

Methods should be briefly mentioned in the abstract.

> they are mentioned: Forty in-depth semi-structured interviews were performed.

Also add conclusion and future recommendation of the study in the abstract.

>added

Line 19-20 the sentence must be revised not clear grammatically.

>we rephrased the abstract

Line 25 change areas into area.

>done

Line 29-31 the sentences are discussion which should not be add in the abstract.

> There is already some discussion but we added another sentence.

Line 42 should be cited with relevant study.
DOI: 10.56042/ijtk.v21i3.31454

> now we cited this paper

Line 45 must be cited with other relevant studies. The following studies could be helpful. https://doi.org/10.1016/j.chnaes.2021.08.002

> we think we provided enough references in this part of intro

In the introduction section line 85-89 should be cited with relevant study.

>we rearrange the paragraphs and changed the position of references as they were already in the text.

There is no need of Figure 2

> we removed figure 2

Add future recommendations in the conclusion.

>added

 The author should revised language and grammar of the study as well.

>The study was originally checked by an experience British copy-editor but we checked the paper again.

Revision 2

Abstract

Line 11 and 15: Please, present clearly the study area.   

>done

Introduction

Line 42: It will be better to say “Some wild food plants

>done

Line 64: probably you mean ethnogenic

>yes, changed

Lines 71-80: needs reference.

>done

Line85-88: needs reference.

>we rearranged the paragraphs and clarified the reference situation

Line 95: east and west what?

 >we removed this nonsensical phrase

Methods

Lines 102-112: Central Lika, Lika-Senj County, Lika county - please, precise administrative/geographical areas.

>done

Line 114: add the reference list number, 26 maybe?? I guess the authors obtained permission for the use of the map?

>we changed the map

Line 120: Add period for the Patriotic war.

>done

Lines 121-2: reduction of the total number of inhabitants = depopulation.

>changed

Line 126-7: Agricultural land is gradually “taken over” by fallow and so-called unorganized or uncultivable land - Please, re-phrase.

>we removed this phrase and rephrased the next sentence to:. The cessation of animal husbandry and the abandonment of pastures and meadows has led to progressive succession, causing pastures to disappear and be replaced by scrubland and eventually forest, which is the climax (final stage) of vegetation in Croatia.

Line 128: … and overgrowth of grasslands… - change to encroachment of thickets or other suitable expression.

>see above for lines 126-127

Lines 171, 209, 212: Please re-check the number of categories (7, 8, 10), 6 in the Results.

>ok, it’s 7

Religion/Ethnicity of the participants should be stated in this section.

>done

Results

Line 244: Please, add authors’ names when species is first mentioned in the text.

>done

Line 246: Probably you mean Figure 3?

yes, corrected to Figure 2 actually

Table 1: Re-check data

            Brassica oleracea, Phaseolus vulgaris  - only MD use?

>of course, people use it for food , thank you.

            Gentiana lutea L. ssp.symphyandra (Murh) Hayek - Gentiana lutea L. ssp.symphyandra (Murb) Hayek

done

            Nicotiana: past? Use

> yes, in the past grown for smoking they don't grow Nicotiana today.

            nutrition - should be replaced with food/feed (fodder) - unify the text in Use

>changed

            Verbena officinalis - used for diarrhea, category FO

>mistake, it’s MD, changed, thank You

Lines 267-8: JI - Jaccard index? - not included in the Methods

>yes, done and the abbreviation clarified                                        

Line 349-50: squeezed cell juice from the leaves 349 of unripe beans - please rephrase.

>done, juice squeezed from raw green bean leaves (Phaseolus vulgaris L.)

Line 352: The authors have to specify if the application of Helleborus dumetorum  is inside the Larynx or on the skin at the outside(skin) surface of this part of the neck located close to the head

>The second, on the skin at the outside(skin) surface of this part of the neck located close to the head

Line 360: Figure 9 is missing in the text.

>we removed the figure

Discussion

Lines 363-70: Please, move to Results, include the ANOVA in the Methods.

>done

It would be more informative if Authors add the number of participants per municipality so to reader to access if the representativeness of the sample supports statements in Lines 379 and thereafter.

>done

Lines 386-91: It will be beneficial to extend the Discussion with comparison with other Balkan countries (e.g.10.3390/plants10112520; 10.1080/03670240600648963; 10.1016/j.hermed.2020.100344, etc.).

>done

Line 402: Juniperus communis is fermented or it is used only for aromatization of rakija?

>no, fermentation (destilation)

Line 408: Oša  is made only with pears?

> A dish called oša consists of a thick compote of dried pears or prunes/fresh plums with homemade bread. However, this tradition has been forgotten in the last 50 years

Line 451: Fix Hellebotus to Helleborus

>done

Reviewer 2 Report

Abstract

Line 11 and 15: Please, present clearly the study area. 

Introduction

Line 42: It will be better to say “Some wild food plants

Line 64: probably you mean ethnogenic

Lines 71-80: needs reference.

Line85-88: needs reference.

Line 95: east and west what?

Methods

Lines 102-112: Central Lika, Lika-Senj County, Lika county - please, precise administrative/geographical areas.

Line 114: add the reference list number, 26 maybe?? I guess the authors obtained permission for the use of the map?

Line 120: Add period for the Patriotic war.

Lines 121-2: reduction of the total number of inhabitants = depopulation.

Line 126-7: Agricultural land is gradually “taken over” by fallow and so-called unorganized or uncultivable land - Please, re-phrase.

Line 128: … and overgrowth of grasslands… - change to encroachment of thickets or other suitable expression.

Lines 171, 209, 212: Please re-check the number of categories (7, 8, 10), 6 in the Results.

Religion/Ethnicity of the participants should be stated in this section.

Results

Line 244: Please, add authors’ names when species is first mentioned in the text.

Line 246: Probably you mean Figure 3?

Table 1: Re-check data

            Brassica oleracea, Phaseolus vulgaris  - only MD use?

            Gentiana lutea L. ssp.symphyandra (Murh) Hayek - Gentiana lutea L. ssp.symphyandra (Murb) Hayek

            Nicotiana: past? use

            nutrition - should be replaced with food/feed (fodder) - unify the text in Use

            Verbena officinalis - used for diarrhea, category FO

Lines 267-8: JI - Jaccard index? - not included in the Methods       

Line 349-50: squeezed cell juice from the leaves 349 of unripe beans - please rephrase.

Line 352: The authors have to specify if the application of Helleborus dumetorum  is inside the Larynx or on the skin at the outside(skin) surface of this part of the neck located close to the head

Line 360: Figure 9 is missing in the text.

Discussion

Lines 363-70: Please, move to Results, include the ANOVA in the Methods. It would be more informative if Authors add the number of participants per municipality so to reader to access if the representativeness of the sample supports statements in Lines 379 and thereafter.

Lines 386-91: It will be beneficial to extend the Discussion with comparison with other Balkan countries (e.g.10.3390/plants10112520; 10.1080/03670240600648963; 10.1016/j.hermed.2020.100344, etc.).

Line 402: Juniperus communis is fermented or it is used only for aromatization of rakija?

Line 408: Oša  is made only with pears?

Line 451: Fix Hellebotus to Helleborus

Author Response

Dear Reviewers

we corrected the text to nearly all of your comments (see the answers below)

Revision 2

Abstract

Line 11 and 15: Please, present clearly the study area.   

>done

Introduction

Line 42: It will be better to say “Some wild food plants

>done

Line 64: probably you mean ethnogenic

>yes, changed

Lines 71-80: needs reference.

>done

Line85-88: needs reference.

>we rearranged the paragraphs and clarified the reference situation

Line 95: east and west what?

 >we removed this nonsensical phrase

Methods

Lines 102-112: Central Lika, Lika-Senj County, Lika county - please, precise administrative/geographical areas.

>done

Line 114: add the reference list number, 26 maybe?? I guess the authors obtained permission for the use of the map?

>we changed the map

Line 120: Add period for the Patriotic war.

>done

Lines 121-2: reduction of the total number of inhabitants = depopulation.

>changed

Line 126-7: Agricultural land is gradually “taken over” by fallow and so-called unorganized or uncultivable land - Please, re-phrase.

>we removed this phrase and rephrased the next sentence to:. The cessation of animal husbandry and the abandonment of pastures and meadows has led to progressive succession, causing pastures to disappear and be replaced by scrubland and eventually forest, which is the climax (final stage) of vegetation in Croatia.

Line 128: … and overgrowth of grasslands… - change to encroachment of thickets or other suitable expression.

>see above for lines 126-127

Lines 171, 209, 212: Please re-check the number of categories (7, 8, 10), 6 in the Results.

>ok, it’s 7

Religion/Ethnicity of the participants should be stated in this section.

>done

Results

Line 244: Please, add authors’ names when species is first mentioned in the text.

>done

Line 246: Probably you mean Figure 3?

yes, corrected to Figure 2 actually

Table 1: Re-check data

            Brassica oleracea, Phaseolus vulgaris  - only MD use?

>of course, people use it for food , thank you.

            Gentiana lutea L. ssp.symphyandra (Murh) Hayek - Gentiana lutea L. ssp.symphyandra (Murb) Hayek

done

            Nicotiana: past? Use

> yes, in the past grown for smoking they don't grow Nicotiana today.

            nutrition - should be replaced with food/feed (fodder) - unify the text in Use

>changed

            Verbena officinalis - used for diarrhea, category FO

>mistake, it’s MD, changed, thank You

Lines 267-8: JI - Jaccard index? - not included in the Methods

>yes, done and the abbreviation clarified                                        

Line 349-50: squeezed cell juice from the leaves 349 of unripe beans - please rephrase.

>done, juice squeezed from raw green bean leaves (Phaseolus vulgaris L.)

Line 352: The authors have to specify if the application of Helleborus dumetorum  is inside the Larynx or on the skin at the outside(skin) surface of this part of the neck located close to the head

>The second, on the skin at the outside(skin) surface of this part of the neck located close to the head

Line 360: Figure 9 is missing in the text.

>we removed the figure

Discussion

Lines 363-70: Please, move to Results, include the ANOVA in the Methods.

>done

It would be more informative if Authors add the number of participants per municipality so to reader to access if the representativeness of the sample supports statements in Lines 379 and thereafter.

>done

Lines 386-91: It will be beneficial to extend the Discussion with comparison with other Balkan countries (e.g.10.3390/plants10112520; 10.1080/03670240600648963; 10.1016/j.hermed.2020.100344, etc.).

>done

Line 402: Juniperus communis is fermented or it is used only for aromatization of rakija?

>no, fermentation (destilation)

Line 408: Oša  is made only with pears?

> A dish called oša consists of a thick compote of dried pears or prunes/fresh plums with homemade bread. However, this tradition has been forgotten in the last 50 years

Line 451: Fix Hellebotus to Helleborus

>done
